# Ontogenetic Changes in the Feeding Behaviour of *Helicoverpa armigera* Larvae on Pigeonpea (*Cajanus cajan*) Flowers and Pods

**DOI:** 10.3390/plants13050696

**Published:** 2024-02-29

**Authors:** Trevor M. Volp, Myron P. Zalucki, Michael J. Furlong

**Affiliations:** 1Agri-Science Queensland, Department of Agriculture and Fisheries, Toowoomba, QLD 4350, Australia; 2School of the Environment, The University of Queensland, St. Lucia, QLD 4072, Australia; m.zalucki@uq.edu.au

**Keywords:** plasticity, foraging, host-plant resistance, plant defence, herbivory, herbivore

## Abstract

Despite substantial research examining caterpillar–plant interactions, changes in the feeding behaviour of lepidopteran larvae as they develop are poorly understood. In this study, we investigated ontogenetic changes in the behaviour of *Helicoverpa armigera* larvae feeding on reproductive structures of pigeonpea (*Cajanus cajan*). Specifically, we examined the preference for and avoidance of pigeonpea flowers and pods of first, second, third, and fourth instar *H. armigera* larvae. We also conducted a no-choice assay to compare the ability of third and fourth instar larvae to penetrate pigeonpea pod walls, which act as a physical defence against herbivory. When presented with a choice between pigeonpea pods and flowers, different instars behaved differently. First and second instar larvae largely avoided pigeonpea pods, instead feeding on flowers; third instar larvae initially avoided pods, but by 24 h, did not strongly discriminate between the structures; and fourth instars demonstrated a preference for pods. When initially placed on pods, first instars were slower than other instars to leave these structures, despite pods being suboptimal feeding sites for small caterpillars. We identified a clear instar-specific ability to penetrate through the pod wall to reach the seeds. Most third instar larvae were unable to penetrate the pod wall, whereas most fourth instars succeeded. Third instars suffered a physiological cost (measured by relative growth rate) when boring through the pod wall, which was not observed in fourth instars. Our study further illuminates the insect–plant interactions of the *H. armigera*–pigeonpea system and provides evidence for the significant changes in feeding behaviour that may occur during lepidopteran larval development.

## 1. Introduction

Holometabolous herbivorous insects may substantially change their feeding behaviour throughout immature development [1]. Some 17.6% of 1137 species of British Lepidoptera make a single marked change in their feeding habit during their larval stages [2]. Such changes may occur due to the quantity and quality of food available, to reduce the risk of mortality (from either lower or higher trophic levels), or because of a larger mandible/body size that enables feeding in different locations and/or on different plant parts.

Lepidopteran larvae are exposed to heterogenous environments (e.g., Figure 1) wherein they should select feeding sites that increase performance and survival, while avoiding sites that decrease performance and increase mortality risk. Whether a feeding site is optimal depends on context—what feeding sites are available—and the instar stage of the larva. As larvae develop, they may switch to feeding in different locations [3], on different types of plant structures [4,5,6], or even to different species of plant [7,8].

In this study, we examine the larval behaviour of *Helicoverpa armigera* (Lepidoptera: Noctuidae), a polyphagous pestiferous moth that attacks numerous plant species throughout its nearly global distribution [9,10,11]. The behaviour of early instar *H. armigera* larvae has been examined in detail—neonates respond to gravity and light, moving up plants and preferring to feed at plant reproductive structures [3,12,13]. As larvae develop, their feeding behaviour changes. On mungbean (*Vigna radiata*), *H. armigera* neonates spend more time searching and located at the plant apex, whereas third instars spend more time feeding and less time at the plant apex [3]. On an artificial diet, third instars move more frequently among diet cubes than neonates [14]. As *H. armigera* larvae develop into larger instars and approach pupation, their food consumption drastically increases [15].

*Helicoverpa armigera* is the most significant insect pest of the major legume crop pigeonpea (*Cajanus cajan*) (Figure 1) [16,17]. Moths are highly attracted to flowering pigeonpea [18,19]. Larvae may feed on pigeonpea plants throughout vegetative and reproductive plant phenological stages [13], but when provided with a choice, early instars avoid leaves and prefer to feed on flowers [18,19,20,21]. As larvae develop into larger instars (Figure 2), they ‘switch’ to feeding on pods [20,22].

In this study, we investigate the ontogenetic changes in the feeding behaviour of *H. armigera* larvae on reproductive structures of pigeonpea and examine the purported ‘switch’ in feeding behaviour of *H. armigera* from flowers to pods. Specifically, we investigate the following questions: (i) do *H. armigera* larvae of different instars prefer to feed on different plant structures, (ii) do early instar *H. armigera* larvae avoid feeding on pods, (iii) does the pod wall limit the ability of early instar larvae to feed on pigeonpea seeds, and (iv) does boring through the pod wall impose a physiological ‘cost’ to larvae?

**Figure 1 plants-13-00696-f001:**
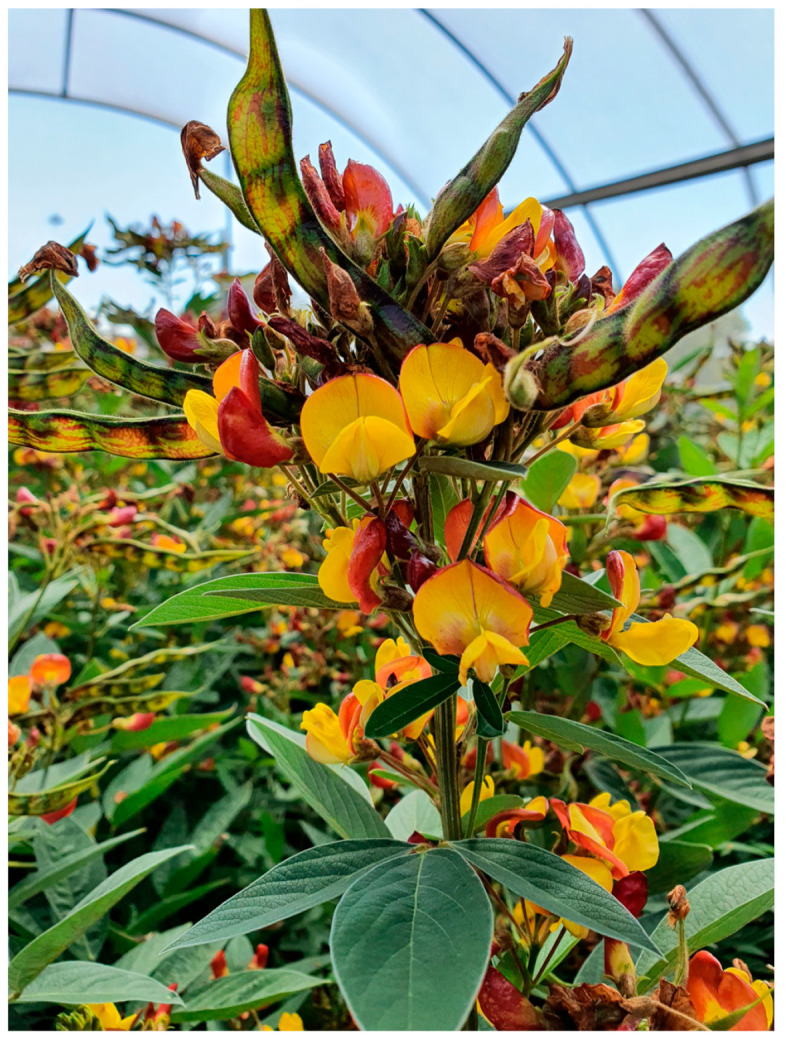
Heterogenous environment of flowering-podding pigeonpea plants (cv. ICPL 86012), where *H. armigera* larvae must make foraging ‘decisions’, see [21] for details on pigeonpea reproductive structures.

**Figure 2 plants-13-00696-f002:**
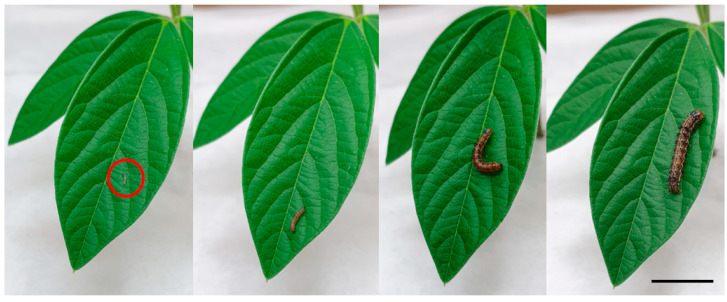
First, second, third, and fourth instar *H. armigera* larvae on the same pigeonpea leaf. After feeding on pigeonpea flowers in the laboratory at 25 °C, it takes 6–7 days for larvae to transition from the first instar to the fourth instar. The red circle indicates the first instar larva and the scale bar = 1 cm.

## 2. Results

### 2.1. Larval Preference for Pods or Flowers

In the larval preference experiment, larval instars differed in their distribution between plant structures at 6 h (χ^2^ = 20.44, df = 3, *p* < 0.001) (Figure 3). By 6 h, the majority of second (93%) and third instar (90%) larvae were on flowers. However, first instars were evenly divided between flowers (56%) and pods (43%), as were fourth instars (flowers 50%; pods 43%). All larvae were located on a plant structure at 6 h, except for two fourth instars which were located on the filter paper.

At 24 h, the larval instar again affected location (χ^2^ = 56.04, df = 3, *p* < 0.001) (Figure 4), with 80% of first instars and all second instars located on flowers. Third instars were evenly distributed between flowers (43%) and pods (53%), but most (83%) fourth instars were found on pods. At 24 h, only three larvae (two fourth instars and a single third instar) were located on the filter paper rather than on the plant structures.

Forty percent of the fourth instar larva had eaten >50% of the flower material presented. Only one third instar consumed this level of flower tissue, but none of the younger instars did. However, when these replicates were removed from the dataset, there was still a significant effect of larval instar on the final larval location (χ^2^ = 46.96, df = 3, *p* < 0.001).

We recorded the presence of visible feeding damage on plant structures to determine which plant structures that larvae fed on. For first instars, it was difficult to determine signs of feeding compared to the older instars; therefore, we excluded them from the analysis. Larval instar influenced whether feeding damage was recorded on flowers only, pods only, or both structures (χ^2^ = 52.28, df = 4, *p* < 0.001). For second instars, 83% of larvae fed only on flowers and 16% fed on both structures. For third instars, 73% of larvae fed on both structures, 16% fed on flowers only, and 10% only fed on pods. For fourth instars, 70% of larvae fed on both structures, 26% fed on pods only, and only a single larva fed on the flower only.

### 2.2. Behavioural Avoidance of Pods

In the pod avoidance experiment, larval instars differed in their distribution between plant structures at 6 h (χ^2^ = 18.53, df = 3, *p* < 0.001) (Figure 5). At 6 h, most first instars (80%) and most fourth instars (63%) were still on the pods, but only 46% of second instars and 26% third instars remained on the pods. At 6 h, two larvae (one third instar and one fourth instar) had left the pod and were found on the filter paper.

At 24 h, larval location was again influenced by larval instar (χ^2^ = 31.40, df = 3, *p* < 0.001) (Figure 6). First instars were evenly distributed between flowers (43%) and pods (43%), with the remainder (14%) located on neither structure. Second instars showed a strong preference for flowers (80%). Third instars were evenly distributed between pods (46%) and flowers (53%), while most fourth instars (86%) were found on pods. There were also two fourth instar larvae located on neither a flower nor a pod. Thirty percent of fourth instar larvae had eaten >50% of flower tissue; when these data are excluded, the distribution of larvae as a function of instar remained significant (χ^2^ = 22.59, df = 3, *p* < 0.001). In this experiment, there was a single dead larva—a first instar that was dead in the pod exudate.

As in the first experiment, we recorded the presence of visible feeding damage on plant structures and excluded first instars due to the difficulty of detecting their feeding damage. Larval instar influenced whether feeding damage was recorded on flowers only, pods only, or both structures (χ^2^ = 22.84, df = 4, *p* < 0.001). For second instars, most larvae fed on flowers only (50%), followed by both flowers and pods (27%), and then pods only (23%). For third instars, most larvae fed on both flowers and pods (60%), followed by pods only (20%) and flowers only (20%). For fourth instars, more larvae fed on both flowers and pods (70%) than pods only (30%).

In the pod avoidance experiment, larval instar affected the probability that a larva penetrated the pod through to the seed (χ^2^ = 78.9, df = 3, *p* < 0.001) (Figure 7). Most fourth instars (96%) but only a minority of third (30%) and second instars (6%) penetrated through to the seed. No first instars penetrated the pod through to the seed in this assay.

### 2.3. Pod Feeding of Older Instars

Larval instar and pod treatment influenced whether larvae fed on pigeonpea seeds (χ^2^ = 63.69, df = 3, *p* < 0.001) (Figure 8). All larvae in the open pod treatment fed on seeds, but in the intact pod treatment, only 26% of third instars chewed their way through to the seed, compared with 83% of fourth instars. This result was reflected in the counts of the different types of holes on pods. When analysing the intact pod treatment only, the larval instar influenced the number of wall holes (F_1,58_ = 10.92, *p* = 0.0016) and seed holes (F_1,58_ = 28.06, *p* < 0.001) (Figure 9). Third instar larvae created more wall holes but fewer seed holes than fourth instars. Again, when only analysing the intact treatment, fourth instar larvae created larger seed holes than third instars (F_1,30_ = 26.88, *p* < 0.001; Appendix A), but wall hole size did not differ between instars (F_1,37_ = 0.037, *p* = 0.85; Appendix A).

When analysing only the intact treatment larvae (i.e., those which had to chew through the pod wall to feed on seed), the initial larval weight affected the probability that third instars bored through the pod wall to feed on seeds (Z = 2.82, df = 30, *p* = 0.0047), but the larval weight did not affect the probability that fourth instars were able to access the seed (Z = 1.53, df = 28, *p* = 0.13) (Figure 10).

Larval RGR was not affected by instar (F_1,116_ = 1.99, *p* = 0.16), but it was affected by whether pods were intact or open (F_1,116_ = 34.15, *p* < 0.001), and there was also a significant interaction effect between larval instar and pod treatment (F_1,116_ = 26.37, *p* < 0.001).

We investigated if there was an instar-specific cost associated with larvae boring through the pod wall by separating larvae from the intact treatment into two groups—those that penetrated through to the seed (labelled ‘P’ in Figure 11) and those that did not (labelled ‘NP’ in Figure 11). Larval RGR was not affected by instar (F_1,114_ = 2.44, *p* = 0.121), but it was significantly affected by pod treatment (F_1,114_ = 47.18, *p* < 0.001) and there was a significant interaction effect between larval instar and pod treatment (F_2,114_ = 4.05, *p* = 0.02). The RGR of third instar larvae that had to bore through the pod wall (3rd intact *p*) was significantly lower that the RGR of those which did not (3rd open) (Fishers LSD, *p* <0.05), but there was no difference between the larval RGR of fourth instars that had to bore through the pod wall (4th intact P) and those which did not (4th open) (Figure 11).

## 3. Discussion

In this study, we examined ontogenetic changes in the feeding behaviour of *H. armigera* larvae feeding on pigeonpea flowers and pods. Using simple laboratory assays, we identified clear differences among larval instars in their preference for, avoidance of, and ability to feed on different pigeonpea reproductive structures. When offered a choice of feeding sites; early instars preferred to feed on flowers and older instars preferred to feed on pods. Smaller instars are limited in their ability to penetrate though pigeonpea pod walls, preventing them from feeding on pigeonpea seeds. Finally, the process of boring through the pod wall imposes a physiological cost on third instar larvae, but not on fourth instars.

In our larval preference experiment, we found that, over a short time frame (6 h), second and third instar larvae overwhelmingly selected flowers as their feeding sites, whereas first and fourth instars were more evenly distributed between the plant parts offered (Figure 3). It was surprising that first instars were evenly divided between pods and flowers at 6 h, as first instars do not survive well on large pigeonpea pods compared to flowers [19]. However, after 24 h, most first and second instars were found on flowers (Figure 4), whereas third instars were evenly distributed between pods and flowers and most fourth instars were found on pods.

The objective of our pod avoidance experiment was to investigate what would happen if a larva was initially located on a pod (a suboptimal feeding site for smaller instars). First instars were slow to leave the pods compared to older instars (Figure 5). Our results from both assays indicate that second and third instar larvae are more mobile than first instars, likely due to their larger body size enabling a faster crawling speed. Therefore, larger instars may correct for ‘mistakes’ (i.e., choosing a suboptimal feeding site) more quickly than first instars. Pod wall trichomes may also slow the ability of first instars that initially select a pod (or are placed on a pod) to relocate to flowers.

The pod avoidance assay revealed the potential consequences of first instar larvae making an initial ‘poor’ selection of feeding site. At 24 h, only 43% of first instars had relocated to flowers (compared to 80% in the preference assay), and none of the first instars penetrated the pod wall through to the seed (Figure 7). Despite so many first instars remaining on the pods at 24 h, only a single first instar died in this assay after being trapped in pod exudate. First instar larvae may simply require longer than 24 h to relocate by crawling to a more suitable site. It is worth noting that the Petri dish assays prevented first instars dispersing by silking, which is an important dispersal mode for small larvae [23]. Identifying the mechanisms by which larvae of different instars disperse from poor feeding sites (crawling vs. silking) warrants further investigation.

Taken together, our Petri dish assay results indicate that first and second instars prefer feeding on flowers and tend to avoid pods, third instars are more evenly distributed between the structures, and fourth instar larvae prefer to feed on pods. This indicates that the third instar is the developmental stage at which *H. armigera* larvae may ‘switch’ to feeding on large filling pods.

The walls of pigeonpea pods function as a physical defence that prevents smaller instars from feeding on pigeonpea seeds (Figure 7 and Figure 8). When restricted to pods in a no-choice assay, most fourth instars penetrated the pod through to the seed, but third instars struggled (Figure 8). Third instars repeatedly attempted to penetrate the pod wall, as evidenced by the larger number of ‘wall holes’ (Figure 9). Third instars also suffered a physiological cost of pod boring (evidenced by a reduced larval RGR) not documented for fourth instar larvae (Figure 11). This cost is likely due to the increased time and physical activity required for third instars to make their way through the pod wall. We suspect that a larger mandible size enables fourth instars to penetrate the pod wall more easily, resulting in the larger seed holes made by fourth instars (Appendix A). However, larval weight may also play a role (Figure 10), with a larger body mass potentially providing larvae with more energy reserves to persevere in chewing to penetrate the pod wall.

Ultimately, our experiments placed *H. armigera* larvae in artificial conditions to simplify the context in which they make foraging decisions. On whole pigeonpea plants, the foraging environment is more heterogenous (Figure 1) and it is important we place our results into the context of our knowledge of the *H. armigera*–pigeonpea system. As we have shown elsewhere [19,21], *H. armigera* moths are highly attracted to flowering pigeonpea plants and lay most of their eggs on flowers. Although eggs may be laid at other crop stages, floral structures are important for the establishment and development of early instar larvae, which preferentially feed at these sites. As larvae develop, flowering pigeonpea plants develop contemporaneously. Other studies have indicated that larvae ‘switch’ feeding from flowers to pods [20,22]. However, rather than a ‘switch’, this phenomenon might be better thought of as an increased capacity of *H. armigera* larvae to feed on a range of structures as they develop. Fourth instar larvae will still feed on flowers (Figure 3 and Figure 5), although in our assays, after 24 h, they did prefer pods (Figure 4 and Figure 6). Future work investigating the foraging of *H. armigera* larvae of different instars on whole pigeonpea plants would likely prove useful.

In a set of simple laboratory assays, we have demonstrated large changes between instars in the feeding behaviour of *H. armigera* larvae. In our experimental setup, it takes 6–7 days at 25 °C for a moderately susceptible *H. armigera* neonate to develop into a fourth instar larva that can feed on large pigeonpea pods ‘cost-free’ and cause seed damage which plants are likely unable to compensate for. Development from the third to fourth instar drastically changes the ability of larvae to feed inside large-filling pigeonpea pods. The challenge for pest management researchers, therefore, is to design strategies that increase the mortality of the cryptic small instar larvae before they quickly reach the damaging large instars (which are also more tolerant to insecticides and biopesticides).

Ontogenetic changes in the ecology of lepidopteran larvae have been underexplored due to understandable experimental difficulties. However, studying how ontogeny influences interactions among caterpillars and other trophic levels is a major frontier in lepidopteran ecology [24]. The interactions between larval instar, plant factors (phenology, structure availability, constitutive and induced defences), natural enemies (predators, parasitoids, and pathogens), and larval nutrient regulation are frighteningly complex. We refrain from speculating what methods will be more useful moving forward—the current reductionist approach disentangling the various factors to examine their effect (e.g., this study), or perhaps novel approaches using high-throughput data collection (e.g., videography, time lapse photography, etc.). However, if we desire to understand larval behaviour as a function of ontogeny, we must evaluate the best approach to generate and test hypotheses that include not only plant traits, but also nutritional geometry and natural enemies.

## 4. Materials and Methods

### 4.1. Plants

To obtain the relevant reproductive structures (flowers and large filling pods) for our assays, we grew pigeonpea plants of a short duration, determinate pigeonpea cultivar (ICPL 86012), in a controlled-temperature glasshouse (27 °C day, 25 °C night). Seeds were planted in 200 mm (4 L) ANOVA^TM^ pots using a 2:1 mix of commercial potting mix (Premium Potting Mix, Searles, Kilcoy, Australia) and sand. Plants were watered regularly, as required, and no additional fertiliser was provided. Plants were not treated with any insecticides, and any glasshouse pests were physically removed upon detection. Under these growing conditions, ICPL 86012 plants reached flowering in approximately 8 weeks. For our assays, we sourced flowers from plants that were 8–10 weeks old and large filling pods from plants that were 10–12 weeks old.

### 4.2. Insects

*Helicoverpa armigera* moths and larvae were obtained from a laboratory culture maintained at the Queensland Department of Agriculture and Fisheries laboratory in Toowoomba, Australia. The culture was established from insects collected from various field crops from south-east Queensland, Australia, in 2020 and the colony was regularly supplemented with field-collected insects to minimise inbreeding. Moths were kept in 5 litre plastic buckets and supplied with 10% sucrose solution using a cotton wick in a 70 mL plastic container. An 18 cm hole was cut in the bucket lid and the edges of the lid were used to secure nappy liner (bamboo rayon), which was used as an oviposition substrate. Eggs were removed daily, washed in 1% sodium hypochlorite solution, and collected onto filter paper using vacuum filtration. The filter paper was allowed to air dry, then placed in 90 mm Petri dishes with the edges sealed with parafilm until the neonates hatched. Upon hatching, neonate larvae were placed in groups onto a soybean-flour-based artificial diet (recipe modified from [25], ingredients provided in [21]) in 500 mL rectangular plastic containers. When larvae developed to the third instar, they were transferred to fresh diet in 32-well plastic trays, where they remained until pupation. Pupae were washed in 1% sodium hypochlorite, air-dried, and placed in 500 mL containers until eclosion. The colony and all experiments were maintained in a controlled temperature room (25 ± 2 °C, 12:12 L:D).

The larvae used in behavioural assays were fed on pigeonpea flowers, mimicking the natural feeding progression of larvae, to prevent any confounding effects of switching larvae from an artificial diet to plant parts [26]. Neonates were individually placed in a 90 mm Petri dish containing filter paper moistened with distilled water and provided with a single pigeonpea flower. To obtain larvae of different instars, old flowers were replaced with fresh flowers at days 4, 6, 7, and 8. Using this method, larvae typically reached the second instar in 2–3 days, third instar in 4–5 days, and fourth instar in 6–7 days. We established different larval cohorts on consecutive days to provide us with different instars to compare simultaneously in our assays. Larvae in the first instar (<4 h old) treatment group were not fed before their use in assays and all larger larvae were starved for 4 h before they were used in assays.

### 4.3. Larval Preference for Pods or Flowers

In the first experiment, we examined preference for flowers versus pods for first, second, third, and fourth instar *H. armigera* larvae. The experiment was conducted in 90 mm Petri dishes lined with water-moistened filter paper. In each dish, we placed one pod (large filling stage) and one flower on either side of the Petri dish (separated by approximately 2 cm). We randomised which side of the Petri dish either structure was placed. We then placed a single larva on the centre of filter paper, equidistant from either plant structure. Petri dishes were sealed with parafilm and placed in the controlled-temperature room. We recorded the location of larvae every hour for the first 6 h, and then again at 24 h. At 24 h, we recorded the presence of visible feeding damage on either plant structure. Thirty replicates were conducted for each instar.

### 4.4. Behavioural Avoidance of Pods

In the second experiment, we examined if different instar *H. armigera* larvae (first, second, third, and fourth instars) avoid feeding on pods in a Petri dish assay. Assays were conducted in a similar manner to the preference assay, except at the start of the assay, where we placed all larvae on pods (mimicking an initial feeding choice of pod). We monitored the location of larvae hourly for the first 6 h, and then again at 24 h. At 24 h, we recorded whether there was visible damage to either plant structure and if larvae had penetrated the pod through to the seed. Thirty replicates were conducted for each instar.

### 4.5. Pod Feeding of Older Instars

In the third experiment, guided by our results from the first two assays, we examined the pod feeding behaviour of third and fourth instar larvae. Specifically, we examined (i) if third and fourth instar larvae were able to feed inside pods under no-choice conditions, (ii) if the pod wall presents a barrier to larvae feeding on seeds, and (iii) if boring through the pod wall imposes a physiological cost on larvae.

Third and fourth instar *H. armigera* larvae were divided into two pod treatments. In the control treatment (‘intact’), larvae were provided access to a single large filling pigeonpea pod. In the second treatment (‘open’), pigeonpea pods were sliced along their ventral seam using a scalpel blade. The seam was then prised open slightly to provide larvae with access to all seeds within the pod without having to chew through the pod wall. Larvae and pods were placed inside 50 mL centrifuge tubes which were positioned vertically to allow larvae to access the entire surface of the pods. As in previous assays, larvae were starved for 4 h before placement into the tubes. After the starvation period, larvae were weighed on a microbalance (HR-250AZ; A&D, Tokyo, Japan) to obtain initial weights. After 24 h, larvae were removed from the tubes and re-weighed. Larval relative growth rates (RGRs) were calculated using Equation (1):RGR = ln(*wt*1) − ln(*wt*0)(1)
where *wt*0 is the initial larval weight after 4 h of starvation and *wt*1 is the larval weight after the 24 h assay. At 24 h, we recorded the number and diameter of holes on the pods. Holes were defined as either ‘wall holes’, where larvae had fed on the pod wall but not penetrated through to the seed, or ‘seed holes’, where larvae had fed through the wall and into the seed. For the hole diameter, we took two linear measurements with Vernier calipers (Protech, Shenzhen, China)—one measurement at 90° to the ventral pod edge and the second at 90° to the first measurement. In this experiment, the third instar treatments were replicated 31 times each and the fourth instar treatments were replicated 29 times each.

### 4.6. Statistical Analyses

We compared the location of larvae and evidence of feeding damage in our Petri dish choice assays using chi-square tests. We compared the pod hole counts and larval relative growth rates from the third experiment using ANOVAs, and post hoc comparisons were conducted using Fisher’s LSD test. We square-root transformed (1 + sqrt(x)) our RGR data to meet ANOVA assumptions. We conducted logistic regression to examine how the larval weight influenced the probability of penetrating the pod walls. All analyses were performed in R version 3.6.2 [27]; for Fisher’s LSD tests, we used the package ‘agricolae’ [28]; and graphs were made with the package ‘ggplot2’ [29].

## Figures and Tables

**Figure 3 plants-13-00696-f003:**
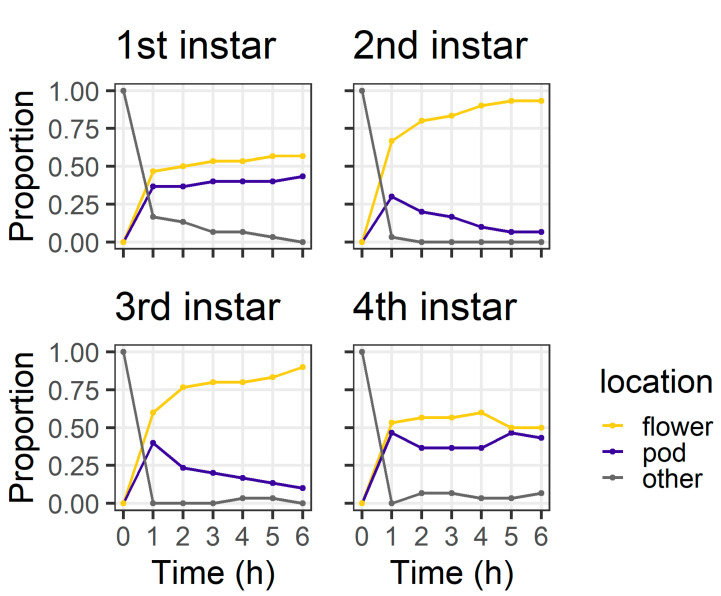
Proportion of larvae at different locations at each of the six hourly observations in the larval preference for pods or flowers experiment; “other” indicates larvae were on the filter paper or the internal walls of the Petri dish (n = 30 for each larval instar).

**Figure 4 plants-13-00696-f004:**
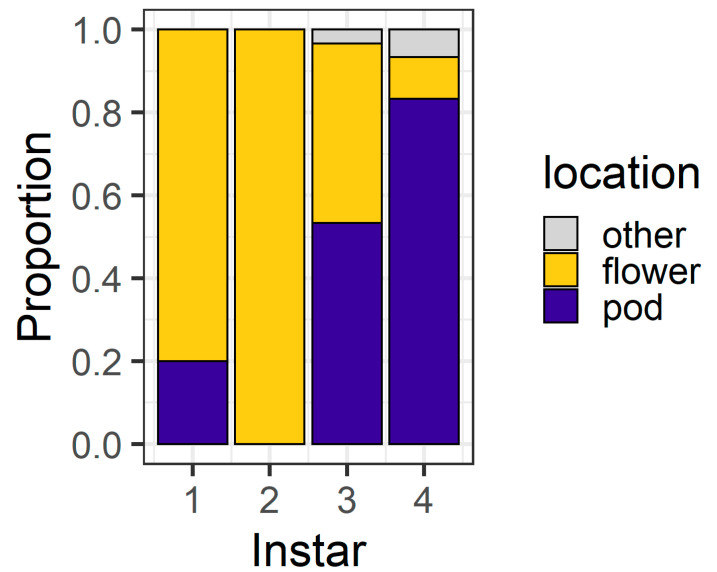
Proportion of larvae at different locations after 24 h in the larval preference for pods or flowers experiment; “other” indicates larvae were on the filter paper or the internal walls of the Petri dish (n = 30 for each larval instar).

**Figure 5 plants-13-00696-f005:**
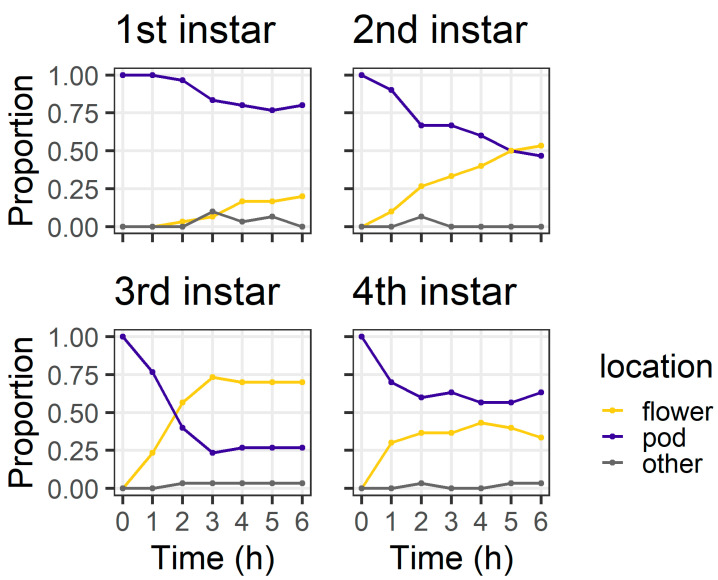
Proportion of larvae at different locations at each of the six hourly observations in the pod avoidance experiment; “other” indicates larvae were on the filter paper or the internal walls of the Petri dish (n = 30 for each larval instar).

**Figure 6 plants-13-00696-f006:**
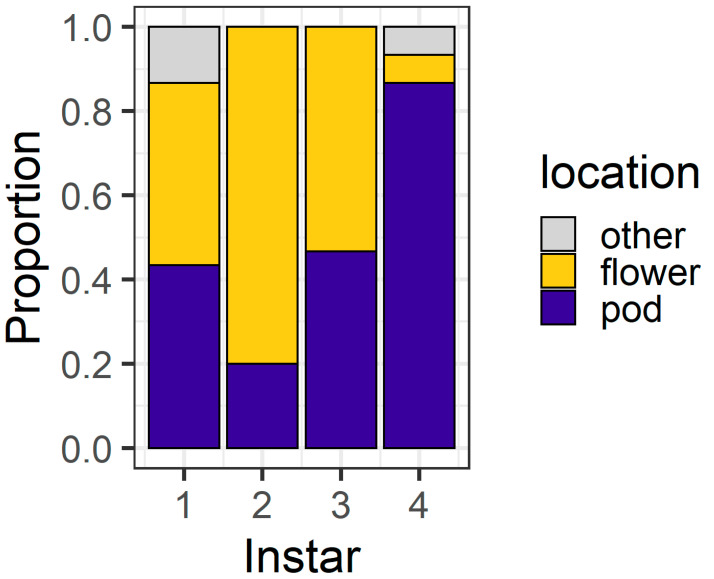
Proportion of larvae at different locations after 24 h in the pod avoidance experiment. Other includes larvae on the filter paper or the internal walls of the Petri dish.

**Figure 7 plants-13-00696-f007:**
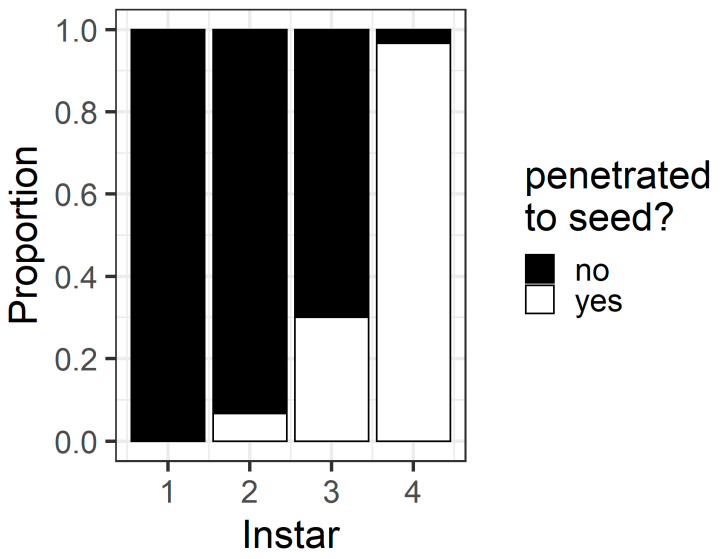
Proportion of larvae of different instars that penetrated the pod wall through to the seed within 24 h in the pod avoidance experiment. The white bars indicate the proportion of larvae that did penetrate through to the seed, and the black bars indicate the proportion of larvae that did not penetrate through to the seed.

**Figure 8 plants-13-00696-f008:**
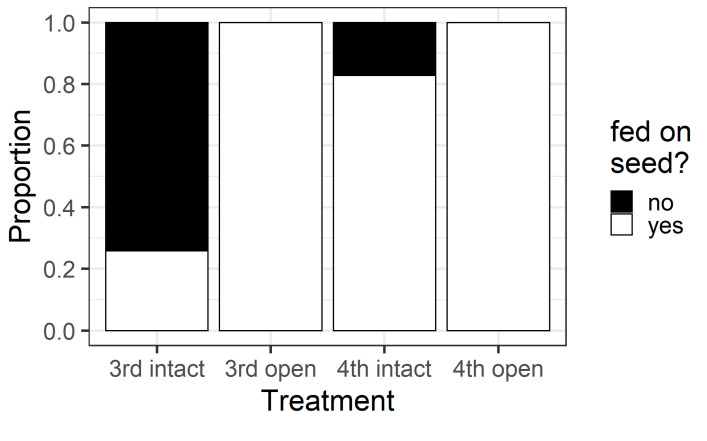
Proportion of larvae of different instars that fed on pigeonpea seeds during the older instar pod feeding experiment. All larvae in the open treatment fed on the seed as they did not have to bore through the pod wall. However, larvae in the intact treatment had to chew through the pod wall to feed on seeds.

**Figure 9 plants-13-00696-f009:**
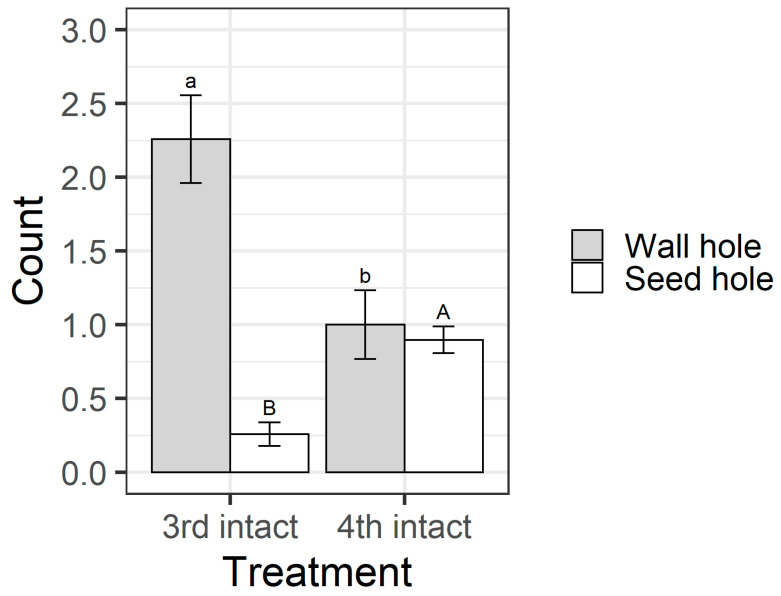
Mean hole counts from the older instar pod feeding experiment. Only data from the intact pod treatment are presented/analysed, as larvae in the open treatment had ready access to seeds. “Wall hole” indicates a larva has fed on the pod wall but not penetrated through to the seed and “seed hole” indicates that the larva has penetrated through to the seed. Bars are the means, and error bars are standard errors. Different letters indicate a significant difference between instars for the same hole type (wall hole = lower case letters and seed hole = upper case) according to Fisher’s LSD test.

**Figure 10 plants-13-00696-f010:**
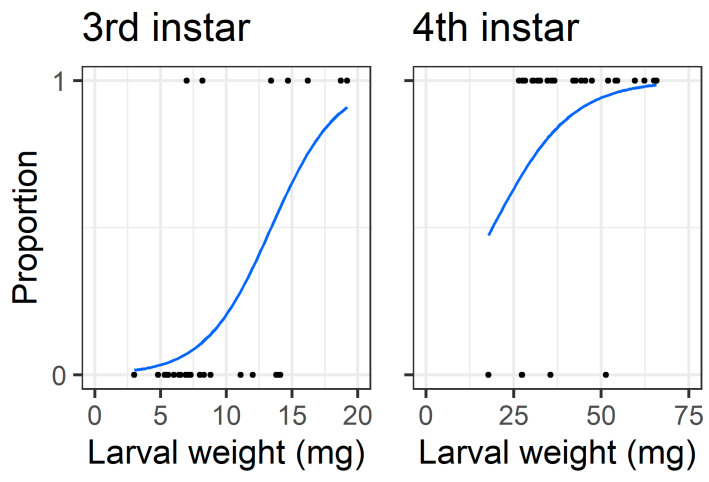
Logistic regressions for feeding on seed in the older instar pod feeding experiment, only for larvae in the control treatment where they had to chew through the pod wall to feed on seed. The y-axis represents success/failure whether a larva managed to feed on seed, 1 = fed on seed and 0 = did not feed on seed. The x-axis is the larval weight at the start of the experiment; note the different scale between third and fourth instars on the x-axis.

**Figure 11 plants-13-00696-f011:**
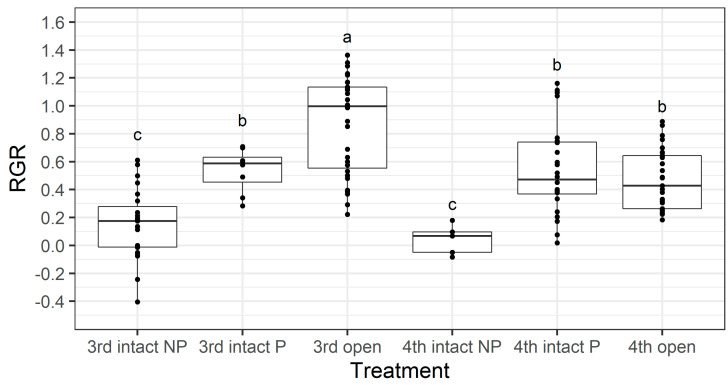
Larval relative growth rate (RGR) from the older instar pod feeding assay. NP = larvae did not penetrate the pod through to the seed, whereas P = larvae penetrated though the pod through to the seed. The box indicates the interquartile range, the horizontal line is the median, and individual data points are shown with black dots. Different letters indicate a significant difference according to Fisher’s LSD test. Untransformed values are presented; however, square-root transformed values were used for analysis.

## Data Availability

Data used in this study are available from the corresponding authors upon request.

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
