# Peer review of "Ontogenetic Changes in the Feeding Behaviour of Helicoverpa armigera Larvae on Pigeonpea (Cajanus cajan) Flowers and Pods"

_plants, 2024, doi:10.3390/plants13050696_

Round 1

Reviewer 1 Report

Comments and Suggestions for Authors
General

The Abstract says it well: “Despite substantial research examining caterpillar-plant interactions, changes in the feeding behaviour of lepidopteran larvae as they develop are poorly understood”.
I agree.

The text is by Native English authors and the data graphs are clear (one exception).
Experiments appear meticulously performed [moths are not my main field, but I have experience in their behavioural ecology in the lab].

The discussion is a bit repetitive

The stats are done in an ‘optimistic way’, not well taking into consideration the conditions/assumptions of the different procedures.

Specifics by row/sections

71      the first Instar image is barely visible on a good colour laser print –too small. I suggest adding a small black arrow.

79ff (p 3)     A heavy reading as 1) data figs on next page only (Tech editor could help!),
many 2) data numbers %’s rows 97-102
and 3) stats outputs. The latter could maybe be put in connection to their graphs.

103, 139    Fig 3 & 5 both spend a lot of grey ink on a background grid and then use the same colour for 1 of 3 lines. Grey on grey does not “help the reader” & the grid itself is of no use. I hope the software just has this grid as default and could easily be omitted? Otherwise, change the software, these graphs are not complex!
All 11 graphs have this redundant grid but is directly disturbing to the reader in Figs 3 &5.

147, 182            Insert, as in Figs 1 -6, a small heading above the category names for explanation such as “Penetration success” (7) and “Fed on seed” (8).

154ff                   more for the Editor: Reading is tough as here is a full page of text (stats etc) relating to 4 graphs but only one (fig 8) is visible.

191-192  How many Standard Deviations, one or two?

195   Which software /package / commands were used for the logistic regression?

 200, 390-385    Fig 11 & section statistical analysis.

Fig 11 shows, apart from biology information, that the data have problems of normality (gaps in the distribution) and correlation of means and variance (high means ó high variation) which jeopardises two major assumptions of parametric stats such as ANOVA. This info will be found in most books of statistics except the very simple ones, but not always in software guides.
Multiple post-hoc test have their own problems. Besides, Fishers LSD (least sign difference) is among the many post-hoc tests one of the most likely to give too much of “significant results” or in other words, it is one of the least conservative multiple post-hoc tests. I suggest not using it.

 390-385   4.2 statistical analysis

This section has too terse a treatment. “Chi-square test” is not sufficient or even correct Pearson? Fischer exact?.
In general,
for any type of analysis of frequencies or otherwise, preferable to refer to software / procedure /command thereof.

For comments on parametric data analysis: See above Fig 11!

218-235,248-253, 270-271 Discussion.

A bit redundant with too mentioning of Results numbers (figs, %s, etc)

239ff      Nice mini-conclusions ending paragraphs!

292-295     not clear to me what is ‘nutritional geometry’ but it sounds good. It sounds as well very much like something that would be a part of ‘plant traits’, at least for an herbivore. 

Reviewer 2 Report

Comments and Suggestions for Authors

                                                      Comments

Volp et al. provided a Research article on “Ontogenetic changes in feeding behaviour of Helicoverpa armigera larvae on pigeonpea (Cajanus cajan) flowers and pods” Though the topic is of current interest, the manuscript needs a minor substantial revision to be published. Overall, the manuscript is well organized. However, the authors may address the following points.

1.     Revise and describe accurately the caption of Figure 3 and Figure 6 and what the figure depicts, and define all elements found in the figure in the figure caption.

2.     Kindly elaborate on the materials and methods (especially heading numbers 4.3 and 4.4 and line numbers 342-361).

3.     Kindly cite the equation 1 used in materials and methods.

4.     Authors may improve the discussion slide to highlight the outcome of this study and propose future applications.

5.     Check all the references according to journal format (Reference numbers 3 and 6, line numbers 422-223 and 428-430).

6.     Italicize all the botanical names.

7.     Overall editing of English language is required.

Comments on the Quality of English Language

                                                      Comments

Volp et al. provided a Research article on “Ontogenetic changes in feeding behaviour of Helicoverpa armigera larvae on pigeonpea (Cajanus cajan) flowers and pods” Though the topic is of current interest, the manuscript needs a minor substantial revision to be published. Overall, the manuscript is well organized. However, the authors may address the following points.

1.     Revise and describe accurately the caption of Figure 3 and Figure 6 and what the figure depicts, and define all elements found in the figure in the figure caption.

2.     Kindly elaborate on the materials and methods (especially heading numbers 4.3 and 4.4 and line numbers 342-361).

3.     Kindly cite the equation 1 used in materials and methods.

4.     Authors may improve the discussion slide to highlight the outcome of this study and propose future applications.

5.     Check all the references according to journal format (Reference numbers 3 and 6, line numbers 422-223 and 428-430).

6.     Italicize all the botanical names.

7.     Overall editing of English language is required.

Round 2

Reviewer 2 Report

Comments and Suggestions for Authors

Accept in the present form.